# The Influences of Intergenerational Care on Life Satisfaction in Older Adults: Chain Mediation by Children’s Emotional Support and Depression

**DOI:** 10.3390/healthcare13111235

**Published:** 2025-05-23

**Authors:** Qianqian Wang, Maiyu Jing, Kan Tian, Shiyu Xie, Xiaoguang Yang

**Affiliations:** 1School of Elderly Care Services and Management, Nanjing University of Chinese Medicine, Nanjing 210023, China; 20221957@njucm.edu.cn (Q.W.); tiankan@njucm.edu.cn (K.T.); 2School of Health Economics and Management, Nanjing University of Chinese Medicine, Nanjing 210023, China; 20221028@njucm.edu.cn; 3Chinese Hospital Development Institute, Shanghai Jiaotong University School of Medicine, Shanghai 200025, China

**Keywords:** older adults, intergenerational care, life satisfaction, children’s emotional support, depression

## Abstract

**Objective:** To explore the relationship between intergenerational care and life satisfaction of older adults, and to analyze the chain mediating effect of children’s emotional support and depression in this relationship, so as to provide scientific reference for improving the quality of life of older adults. **Methods:** In total, 2970 older adults ≥60 years old from the China Health and Retirement Longitudinal Study (CHARLS) were selected as the study subjects. The process plug-in of SPSS was used, and the chain mediating effect test was carried out following the Bootstrap method. **Results:** Intergenerational care was positively correlated with children’s emotional support and life satisfaction (*r* = 0.123, 0.141, *p* < 0.001) and negatively correlated with depression (*r* = −0.096, *p* < 0.001). The mediating effects of children’s emotional support and depression were significant between intergenerational care and life satisfaction, with a mediating effect of 0.023 (95% *CI*: 0.015–0.033), 0.028 (95% *CI*: 0.014–0.043), and the chained mediating effect of children’s emotional support-depression was also significant, with a mediating effect of 0.006 (95% *CI*: 0.004–0.008). The total indirect effect of children’s emotional support and depression between intergenerational care and life satisfaction was 0.057, accounting for 26.03% of the total effect. **Conclusions:** Intergenerational care not only directly affects life satisfaction of older adults, but also indirectly affects life satisfaction through the independent mediating effect of children’s emotional support and depression, as well as the chain mediating effect of children’s emotional support-depression. It is essential to create a positive and inclusive social environment for the intergenerational care of older adults.

## 1. Introduction

Intergenerational care, defined as grandparents’ active involvement in the raising and daily care of grandchildren, is deeply rooted in China’s Confucian tradition [1,2]. This phenomenon emerges as both a product of evolving social structures in modern China and a manifestation of contemporary intergenerational solidarity, reflecting the continuity and reinterpretation of filial piety in Chinese culture. Confucianism prioritizes “benevolence” (rén) as its ethical core, with “filial piety” (xiào) serving as a pivotal expression of “benevolence”. Filial norms have historically shaped the Chinese worldview of “family-state isomorphism”, wherein familial and societal hierarchies mirror one another [3]. The enduring ethos of “respecting older adult as one’s own kin and cherishing the young as one’s own children” has fostered an implicit intergenerational contract in modern households [4]. This contract operates through reciprocal obligations: grandparents fulfill child-rearing duties, parents uphold filial responsibilities in older adult care, and grandparents reciprocate through intergenerational care. Collectively, these practices sustain a tripartite “nurture–support–renurture” intergenerational cycle at the familial level [5,6]. The traditional ideal of “enjoying life’s sweetness while dandling grandchildren” (hán yí nòng sūn) materializes in this framework, allowing grandparents to derive emotional fulfillment from caregiving, alleviating parental childcare burdens, and reinforcing bidirectional intergenerational solidarity. Such reciprocity critically underpins family cohesion in contemporary Chinese society [7,8,9].

As China’s aging population continues to increase—evidenced by the Seventh National Population Census which shows that 18.7% of the population is aged 60 or older [10]—the social function of intergenerational care has transcended traditional family boundaries, becoming a strategic resource for addressing aging-related challenges. The 14th Five-Year Plan for National Economic and Social Development and the Long-Range Objectives Through 2035 of the People’s Republic of China explicitly proposes “implementing a national strategy to actively respond to population aging”, advocating for the development of older adult human resources [11]. In this context, intergenerational care is regarded as an innovative pathway to activate social participation among older adults while alleviating pressure on the older adult care system [12,13]. Empirical studies indicate that older adults engaged in a caregiving role gain identity through childcare responsibilities, reporting significantly higher life satisfaction than non-caregivers (β = 0.031, *p* < 0.01) [14,15,16], with a reduction in the incidence of depressive symptoms [17,18,19].

The chain mediation model extends traditional mediation analysis by proposing that an independent variable affects a dependent variable through sequential mediators in a temporally or logically ordered pathway [20]. Rooted in Baron and Kenny’s linear mediation framework (X → M → Y), this model emerged to address the limitations of single-mediator paradigms in capturing psychosocial processes characterized by cascading mechanisms [21]. While the early mediation theory conceptualized unidirectional causality, it inadequately explained complex dynamics where variables interact bidirectionally or hierarchically. Preacher and Hayes’ bootstrapping procedures resolved these constraints by enabling non-parametric quantification of indirect effects across multi-mediator chains (X → M1 → M2 → Y) [22]. The model’s theoretical validity is further reinforced by Developmental Systems Theory [23], which asserts that psychosocial constructs, such as familial caregiving and emotional well-being, operate through temporally ordered interactions. Thus, the chain mediation framework bridges methodological precision with the dynamic hierarchies inherent in human behavioral processes.

In light of these considerations, this study innovatively constructs a chain mediation model to explore the transmission mechanisms of emotional support from adult children and depressive symptoms in the relationship between intergenerational care and life satisfaction among older adults. Data from the 2018 China Health and Retirement Longitudinal Study (CHARLS) will be used to empirically validate the chain pathway of “intergenerational care → children’s emotional support → depressive symptoms → life satisfaction”, revealing the dynamic “responsibility-reward” equilibrium within Chinese intergenerational family interactions. The findings provide a theoretical basis for improving family friendly aging policies and advancing the goals of healthy aging strategies.

## 2. Objects and Methods

### 2.1. Research Subjects

The data utilized in this study were derived from the China Health and Retirement Longitudinal Study (CHARLS) conducted in 2018. CHARLS, hosted by the National School of Development at Peking University, is a large-scale interdisciplinary survey project that encompasses samples from 28 provinces (autonomous regions and municipalities) across China, covering 150 county-level units and 450 village-level units. This research focused on older adults aged 60 years and above as the subjects of investigation. Due to the absence of intergenerational care variables in the data collected in 2020, we selected the survey data from 2018 for our analysis. After excluding cases with missing values for key variables, a final sample size of 2970 was obtained. The specific screening process is illustrated in Figure 1.

### 2.2. Variable Selection

For this study, life satisfaction of older adults was selected as the dependent variable, intergenerational care as the independent variable, and adult children’s emotional support and depression as mediating variables [24,25]. The specific explanations of each variable are provided below, with their corresponding assignments detailed in Table 1.

#### 2.2.1. Independent Variable: Intergenerational Care

The independent variable, intergenerational care, was constructed as a binary variable based on responses to the question in the 2018 CHARLS survey: “In the past year, did you or your spouse spend time caring for your grandchildren?”. Respondents answering “no” were coded as 0, while those answering “yes” were coded as 1.

#### 2.2.2. Dependent Variable: Life Satisfaction

Life satisfaction was measured using responses to the question “Overall, how satisfied are you with your life?” in the 2018 CHARLS dataset. Responses were categorized and scored as follows: “not at all satisfied = 0”, “not very satisfied = 1”, “moderately satisfied = 2”, “very satisfied = 3” and “extremely satisfied = 4”, yielding a range of 0–4.

#### 2.2.3. First Mediating Variable: Children’s Emotional Support

The first mediating variable was children’s emotional support. In the CHARLS 2018 survey questionnaire, this was primarily assessed through two questions, “How often do you see your children when you do not live together?” and “How often do you contact your children via phone, text message, WeChat, letter, or email when you do not live together?” Each question offered ten response options. Based on the questionnaire design, responses indicating “almost never”, “once a year”, “once every six months” or “other” were assigned a value of 1; responses of “once every three months”, “once a month” or “every half month” were assigned a value of 2; while responses such as “once a week”, “2–3 times per week” or “almost daily” were assigned a value of 3 [26]. The average score was then calculated. Scores below 1 were classified as low frequency and assigned a value of 1; scores between 1 and 2 were classified as medium frequency and given a value of 2; scores above 3 were classified as high frequency and assigned a value of 3. The scoring range was from 1 to 3.

#### 2.2.4. Second Mediating Variable: Depression

The second mediating variable was depression. The CHARLS survey in 2018 utilized a simplified version of the Center for Epidemiologic Studies Depression Scale (CES-D) [27], which includes eight items describing negative emotions and two items reflecting positive emotions, totaling ten items. Each item on this scale employed a four-point rating system where the options “<1 day”, “1–2 days”, “3–4 days” and “5–7 days” corresponded to scores of 0, 1, 2 and 3, respectively. Notably, the items “I feel hopeful about the future” and “I am very happy” were scored in reverse fashion. The total score ranged from 0 to 30 points and higher scores indicated greater severity of depression symptoms. Specific items included I worry about small things; I find it hard to concentrate on tasks; I feel downhearted; I think everything takes effort; I feel hopeful about the future; I feel fearful; my sleep is poor; I am very happy; I feel lonely and I believe that I cannot continue my life as before.

#### 2.2.5. Control Variables

This study controlled for age (in years), gender (male = 0, female = 1), education level (primary school and below = 0; middle school = 1; high school = 2; university and above = 3), marital status (married = 0, unmarried = 1), residence (town = 0, rural = 1), self-rated health (very bad = 0; bad = 1; general = 2; good = 3; very good = 4), participation in medical insurance (no medical insurance = 0, medical insurance = 1) and pension insurance (no pension insurance = 0, pension insurance = 1) to account for potential confounding effects. Based on the theoretical framework and empirical evidence, the following hypotheses were formulated.

**H1.** 
*“Intergenerational care” has an impact on “life satisfaction”.*


**H2a.** 
*“Intergenerational care” has an impact on “children’s emotional support”.*


**H2b.** 
*“Children’s emotional support” has an impact on “life satisfaction”.*


**H2c.** 
*In the relationship between “intergenerational care” and “life satisfaction”, “children’s emotional support” is a mediating variable.*


**H3a.** 
*“Intergenerational care” has an impact on “depression”.*


**H3b.** 
*“Depression” has an impact on “life satisfaction”.*


**H3c.** 
*In the relationship between “intergenerational care” and “life satisfaction”, “depression” is a mediating variable.*


**H4.** 
*“Intergenerational care” influences “life satisfaction” through a chain mediation pathway involving “children’s emotional support” and “depression”.*


### 2.3. Statistical Methods

Data screening and cleaning were conducted using Stata 17.0 (StataCorp LLC, College Station, TX, USA) before importing into IBM SPSS 27.0 (IBM Corporation, Armonk, NY, USA) for data analysis. To address potential confounding effects, variables identified as confounders were selected based on statistical criteria, specifically variables showing associations with both the exposure and outcome variables at a significance threshold of *p* < 0.20 in preliminary analyses. These variables were subsequently controlled in multivariate models during the statistical adjustment process. Descriptive statistics were presented with counts (*n*) and percentages (%) for categorical data, non-normally distributed continuous data were described using *M* (*P*_25_, *P*_75_). The comparison of categorical variables between groups was performed using chi-square tests. Partial correlation analysis was employed to examine correlations among primary variables. For the chain mediation analysis, all identified confounders were included as covariates in the PROCESS macro to adjust their effects when estimating indirect pathways. Chain mediation effects were tested utilizing SPSS PROCESS macro version 4.2 with the Bootstrap method by selecting Model 6 and conducting random sampling 5000 times. A two-tailed significance level of α = 0.05 was adopted for all statistical analyses [28,29].

## 3. Results

### 3.1. Basic Information of Research Subjects

This study included a total of 2970 older adults, with an average age of 71.75 ± 6.15 years. Among them, 926 participants (31.2%) were involved in intergenerational caregiving, while 2044 participants (68.8%) did not engage in such caregiving activities. Comparisons between the demographic data and the intergenerational caregiving group revealed significant differences in age, gender, education level, marital status, residence, self-rated health status and participation in pension insurance (*p* < 0.05). When comparing life satisfaction scores across groups, differences were noted in gender, marital status, self-rated health status and participation in pension insurance (*p* < 0.05). Similarly for children’s emotional support compared to other variables, age, education level, marital status, residence and self-rated health showed significant disparities (*p* < 0.05). Regarding depression scores, gender, marital status, residence and self-rated health exhibited statistically significant differences (*p* < 0.05). The results of the analyses of differences are shown in Table 2 and Table 3.

### 3.2. Common Method Bias Test

The Harman single-factor test was conducted to analyze 12 variables. The results indicated that there were five factors with eigenvalues greater than 1. The first factor accounted for 15.64% of the variance, which was below the critical threshold of 40%. Therefore, no significant common method bias issues were identified.

### 3.3. Correlation Analysis of Intergenerational Care, Children’s Emotional Support, Depression and Life Satisfaction

In this study, intergenerational care was found to have a positive correlation with children’s emotional support (r = 0.123, *p* < 0.001) and life satisfaction (r = 0.141, *p* < 0.001), while exhibiting a negative correlation with depression (r = −0.096, *p* < 0.001). Additionally, children’s emotional support demonstrated a negative correlation with depression (r = −0.146, *p* < 0.001) and a positive correlation with life satisfaction (r = 0.177, *p* < 0.001). Furthermore, depression was negatively correlated with life satisfaction (r = −0.335, *p* < 0.001). The results of the correlation analysis results are shown in Table 4.

### 3.4. Mediating Model Regression Analysis of Children’s Emotional Support, Depression, Intergenerational Care and Life Satisfaction

After incorporating control variables into the analysis, the results indicated that intergenerational care significantly positively predicts children’s emotional support (β = 0.210, *p* < 0.001) and life satisfaction (β = 0.162, *p* < 0.001), while negatively predicting depression (β = −0.096, *p* < 0.001). Furthermore, children’s emotional support significantly negatively predicted depression (β = −0.095, *p* < 0.001) and positively predicted life satisfaction (β = 0.111, *p* < 0.001). Additionally, depression significantly negatively predicted life satisfaction (β = −0.288, *p* < 0.001). The results of the regression analysis results are shown in Table 5.

### 3.5. Chain Mediation Pathway Analysis of Children’s Emotional Support and Depression

The results of the chain mediation effect analysis indicated that in the model “Intergenerational Care → Children’s Emotional Support → Depression → Life Satisfaction”, the total effect of intergenerational care on life satisfaction was 0.219. Among this, the direct effect accounted for 0.162, representing 73.97% of the total, while the total indirect effect was 0.057, constituting 26.03%. The confidence intervals (95% CI) for all three indirect effect pathways did not include zero, demonstrating that the mediating effects of children’s emotional support and depression were significant. This suggested that these variables played a partial mediating role in how intergenerational care influenced life satisfaction. The results are presented in Table 6 and Figure 2.

### 3.6. Results of Hypothesis Testing

The empirical findings from the chain mediation model support all proposed hypotheses. First, intergenerational care exhibits a significant positive effect on life satisfaction (H1: β = 0.162, supported). Second, intergenerational care enhances children’s emotional support (H2a: β = 0.210, supported), which in turn, positively predicts life satisfaction (H2b: β = 0.111, supported). Furthermore, children’s emotional support bridges the relationship between intergenerational care and life satisfaction (H2c: indirect effect = 0.023, supported). Third, intergenerational care reduces depressive symptoms (H3a: β = −0.096, supported), and lower depression levels correlate with higher life satisfaction (H3b: β = −0.288, supported). The indirect path through depression is statistically significant (H3c: indirect effect = 0.028, supported). Finally, a chain mediation pathway (H4) is confirmed: intergenerational care enhances life satisfaction by first increasing children’s emotional support and then reducing depression, with an indirect effect of 0.006. These findings highlight dual mediating mechanisms in this relationship. The detailed results are shown in Table 7 and Figure 2.

## 4. Discussion

This study involved older adults aged 60 and above from the 2018 CHARLS to conduct a cross-sectional analysis, exploring the intrinsic relationships and mechanisms of influence among intergenerational care, children’s emotional support, depression and life satisfaction in older adults. According to effect analysis, children’s emotional support and depression played mediating roles in the impact of intergenerational care on life satisfaction among older adults. The indirect effects of these two factors accounted for 10.5% and 12.79% of the total effect, respectively. Furthermore, the “children’s emotional support-depression” pathway also exhibited a chain mediation effect within this context, contributing 2.74% to the overall impact of intergenerational care on life satisfaction in older adults.

### 4.1. The Positive Correlation Between Intergenerational Care and Life Satisfaction of Older Adults

Our findings indicated that the involvement of older adults in intergenerational care positively impacted their life satisfaction, suggesting that caring for grandchildren enhanced the overall well-being of older adults. Existing studies have shown that providing intergenerational care contributed to improving the mental health status of middle-aged and older adults [12]. When grandparents offer support in daily living, educational guidance and health monitoring for their grandchildren, they are better able to integrate into a diverse society with this additional social role, thereby enhancing both their psychological and physical health, which further increases their sense of happiness in life [30].

At the same time, based on role strain theory [31], when older adults face conflicts arising from their social roles, they may experience stress and tension. The limitations of time and energy can lead to changes in their quality of life. Therefore, it is essential for society, families and individuals to recognize the significance of intergenerational carefully. Various societal participants should be mobilized to establish protective mechanisms during the process where grandparents provide nurturing for their grandchildren [32].

To promote positive aging policies [33], it is crucial to enhance both the level and quality of medical services available to older adults while ensuring they maintain the good physical health conditions necessary for engaging in intergenerational care. Additionally, strengthening public childcare services and constructing supporting facilities to provide respite services as well as parenting guidance to older adult caregivers looking after grandchildren [34] is essential. Family members, such as children, should also increase financial support across generations to alleviate the practical pressures associated with caring for grandchildren along with reducing caregiving intensity.

### 4.2. The Mediating Role of Children’s Emotional Support in the Relationship Between Intergenerational Care and Life Satisfaction Among Older Adults

The findings of our study indicated that children’s emotional support had a positive predictive effect on the life satisfaction of older adults. As a manifestation of filial piety cultural values, intergenerational care provided by grandparents is often supported by children’s frequent offline visits and online communications. This not only enhances the life satisfaction of older adults but also contributes to the optimal allocation of family resources. Based on altruistic theory [35], older adults can achieve maximum family benefits through personal self-sacrifice, which in turn, fulfills their sense of happiness.

Relevant research suggested that older adults in China tended to prioritize collective well-being, aiming for familial unity and harmony through means such as intergenerational care [36]. The emotional support received from children serves as positive feedback that can enhance the daily living capabilities of older adults, thereby increasing their overall happiness index. In light of this, adult children should not only provide basic financial support but also adopt more flexible work schedules to offer greater companionship to both aging parents and young children. In the context of China, a reciprocal support system is advocated, namely shared intergenerational resources, such as cooperative childcare fees, housing subsidies or flexible financial transfers, which work in synergy with emotional solidarity. This approach helps cultivate a harmonious and friendly family atmosphere while ensuring the physical and mental health of older individuals [37]. Children’s emotional support serves as the critical bridge through which intergenerational care improves life satisfaction, operating via the sequential pathway: grandparental care creates opportunities for intergenerational interaction → interactions elicit emotional support from adult children → emotional support amplifies older adults’ well-being. Amplifying this mediating effect can transform intergenerational care into a genuine “win-win” bond across generations [38].

### 4.3. The Mediating Role of Depression in the Relationship Between Intergenerational Care and Life Satisfaction Among Older Adults

Our findings revealed that depression mediated the relationship between intergenerational care and life satisfaction, exhibiting a negative predictive effect on the latter. Consistent with prior research [39], providing care for grandchildren reduced the likelihood of depressive symptoms among middle-aged and older adults. Such caregiving addresses emotional voids by enriching retirees’ lifestyles, facilitating a sense of self-fulfillment, while simultaneously strengthening intergenerational bonds with adult children. These interactions alleviate loneliness and apathy, further mitigating depressive symptoms and thereby enhancing life quality perceptions [40]. Furthermore, intergenerational engagement during caregiving, such as play activities, stimulated endorphin secretion through moderate physical exertion, improving emotional states. Academic interactions with grandchildren also exercise older adults’ linguistic and memory-related cognitive functions, delaying cognitive decline and counteracting depression rooted in perceived “uselessness” [41]. Additionally, fulfilling caregiving responsibilities reinforces older adults’ perceived competence and authority, which in turn, elevates subjective well-being and mental health, culminating in heightened life satisfaction [42]. Depression thus functions as an “emotional regulator” between grandparental caregiving and life satisfaction: caregiving indirectly liberates psychological resources by reducing depressive levels, subsequently improving emotional states and life satisfaction [43].

### 4.4. The Chain Mediating Role of Children’s Emotional Support and Depression in the Relationship Between Intergenerational Care and Life Satisfaction Among Older Adults

The findings of our study revealed a sequential mediation pathway through which children’s emotional support and depression jointly linked intergenerational care to older adults’ life satisfaction. Caring for grandchildren serves as a catalyst for emotional exchanges between older adults and their adult children, reducing the likelihood of depressive symptoms. Grounded in Social Support Theory [44], intergenerational care may expand older adults’ social networks by nurturing frequent interactions with children and other family members. A robust social network, as a critical source of social support, enhances their capacity to cope with aging-related challenges and stressors [45]. Concurrently, it reinforces their sense of respect and recognition within familial and societal contexts, which not only bolsters self-fulfillment and mitigates psychological burdens and pessimism but also positively elevates life satisfaction [46]. By assisting adult children in grandchild care, older adults convey care and affection, prompting gratitude and filial obligations from their children. This dynamic creates a harmonious family atmosphere, strengthens intergenerational intimacy through shared caregiving responsibilities and fulfills older adults’ sense of self-worth through children’s acknowledgment. Increased positive evaluations further alleviate depressive symptoms, ultimately enhancing life satisfaction [47,48]. Thus, intergenerational care stimulates adult children’s emotional support, creating a reservoir of emotional resources that reduces depressive risks and improves older adults’ mental health, thereby elevating life satisfaction. Amplifying this chain mediation effect may advance the dual aspirations of “productive aging” and “enjoyable aging” [49].

### 4.5. Some Suggestions

As a vital extension of familial care for older adults rooted in Confucian values of filial piety and intergenerational reciprocity, intergenerational care necessitates systemic policy support. First, establish a rights protection mechanism by legislating social security entitlements for caregivers, including pension credits for the caregiving duration to alleviate welfare sacrifices. For instance, Japan implements child tax deductions under the Income Tax Act and Local Tax Act, offering up to 380,000 Japanese Yen annually per child aged 0 to 18, and provides childcare allowances through the Child Allowance Act, granting 15,000 Japanese Yen monthly for children aged 0 to 3. We can learn from and adapt similar practices. Second, refine economic subsidies. Rural areas may implement caregiving allowances for economically dependent older adults, while urban regions adopt tax incentives to encourage financial compensation. Third, institutionalize respite care services through government-community partnerships to provide temporary relief for intensive caregivers. Lastly, implement urban–rural tailored measures. Urban communities should establish parenting guidance centers, whereas rural areas prioritize healthcare investments to mitigate caregiving-related risks.

Inspired by Confucian ethics emphasizing intergenerational harmony, the sustainability of caregiving relies on a four-tiered “society–community–family–individual” network. Socially, media should promote mutual aid culture, countering the stereotype of caregiving as older adults’ sole duty, while enterprises adopt flexible work policies and caregiving leave. Communally, communities can achieve intergenerational care reciprocity through structured volunteer programs [50]. One such model is establishing “co-care hubs” that pair younger volunteers with older adults: retirees provide childcare support to working parents, while younger participants assist elders with technology literacy and mobility assistance. Famillially, strengthen bidirectional engagement via shared childcare activities and role negotiation, which echoes the teaching that “parental kindness and filial reverence sustain harmony”. Individually, empower older adults through mental health education and boundary-setting skills. This integrated framework transforms caregiving into a cross-generational bond of shared well-being.

### 4.6. Limitations and Future Research Directions

The limitations of this study primarily manifest in three aspects. First, measuring the intensity of intergenerational care has been restricted to binary variables, which may not fully capture the complexity of caregiving dynamics. Second, under the urban–rural dual structure, differentiated mechanisms remain unexplored. Rural areas may experience “high burden-low return” caregiving models due to factors like scarce medical resources and economic dependency between generations. Third, the cross-sectional design may limit insights into how policy changes over time may shape intergenerational care patterns.

In future endeavors, our study will develop multidimensional indicators of intergenerational care to explore its complex pathway mechanisms. Additionally, comparative urban–rural analyses will be conducted to examine the mediating mechanisms of structural inequalities. Furthermore, longitudinal data from the CHARLS will be utilized to track the dynamic interactions between policy shifts and intergenerational care patterns.

## 5. Conclusions

This study validates the mechanisms through which intergenerational care affects older adults’ life satisfaction via a chain mediation model. Intergenerational care directly enhances life satisfaction and operates through two independent mediating pathways. First, intergenerational care significantly increases children’s emotional support, which in turn, positively predicts life satisfaction, with children’s emotional support mediating the relationship between intergenerational care and life satisfaction. Second, intergenerational care reduces depression levels, and lower depression further improves life satisfaction, while depression mediates the link between intergenerational care and life satisfaction. Furthermore, intergenerational care sequentially influences life satisfaction through the chain mediation pathway of “children’s emotional support → alleviation of depression”. The data suggest that strengthened intergenerational interactions and reduced psychological stress jointly explain the positive impact of intergenerational care on older adults’ well-being.

## Figures and Tables

**Figure 1 healthcare-13-01235-f001:**
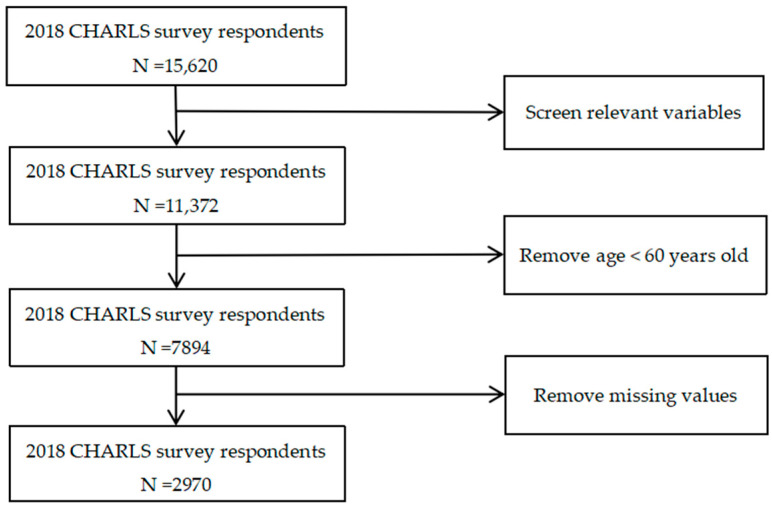
Variable screening flow chart.

**Figure 2 healthcare-13-01235-f002:**
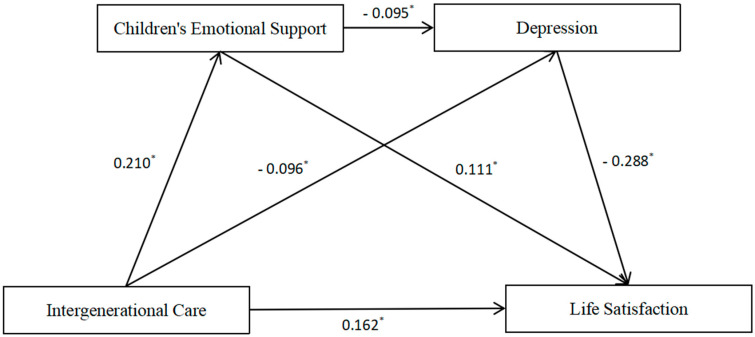
Chain mediation model. **Note:** * *p* < 0.001.

**Table 1 healthcare-13-01235-t001:** Variable assignment.

	Variable	Assignment
Independent Variable	Intergenerational Care	no = 0, yes = 1
Dependent Variable	Life Satisfaction	not at all satisfied = 0
		not very satisfied = 1
		moderately satisfied = 2
		very satisfied = 3
		extremely satisfied = 4
First Mediating Variable	Children’s Emotional Support	almost never, once a year, once every six months, other = 1
		once every three months, once a month, every half month = 2
		once a week, 2–3 times per week, almost daily = 3
Second Mediating Variable	Depression	<1 day = 0, 1–2 days = 1, 3–4 days = 2, 5–7 days = 3 (problems of negative emotions)

		5–7 days = 0, 3–4 days = 1, 1–2 days = 2, <1 day = 3 (problems of positive emotions)

Control Variables	Age	60~69, >69~79, >79
	Gender	male = 0, female = 1
	Education Level	primary school and below = 0
		middle school = 1
		high school = 2
		university and above = 3
	Marital Status	married = 0, unmarried = 1
	Residence	town = 0, rural = 1
	Self-rated Health	very bad = 0
		bad = 1
		general = 2
		good = 3
		very good = 4
	Medical Insurance	no = 0, yes = 1
	Pension Insurance	no = 0, yes = 1

**Table 2 healthcare-13-01235-t002:** Analyses of differences in intergenerational care and life satisfaction (N = 2970).

Variables	Groups	*N* (%)	Intergenerational Care		Life Satisfaction	
			*χ* ^2^	*p*	*χ* ^2^	*p*
Age	60~69	1309 (44.1)	179.268	<0.001	5.644	0.687
	>69~79	1285 (43.3)				
	>79	376 (12.7)				
Gender	Male	1471 (49.5)	5.788	0.016	19.230	<0.001
	Female	1499 (50.5)				
Education Level	Primary school and below	2521 (84.9)	13.741	0.003	3.530	0.060
	Middle school	311 (10.5)				
	High school	115 (3.9)				
	University and above	23 (0.8)				
Marital Status	Married	1785 (60.1)	27.193	<0.001	24.360	<0.001
	Unmarried	1185 (39.9)				
Residence	Town	1059 (35.7)	11.396	<0.001	6.615	0.158
	Rural	1911 (64.3)				
Self-rated Health	Very bad	247 (8.3)	19.685	<0.001	207.396	<0.001
	Bad	822 (27.7)				
	General	1419 (47.8)				
	Good	289 (9.7)				
	Very good	193 (6.5)				
Medical Insurance	No	155 (5.2)	2.199	0.138	9.111	0.058
	Yes	2815 (94.8)				
Pension Insurance	No	369 (12.4)	12.169	<0.001	13.886	0.008
	Yes	2601 (87.6)				

**Table 3 healthcare-13-01235-t003:** Analyses of differences in children’s emotional support and depression (N = 2970).

Variables	Groups	*N* (%)	Children’s Emotional Support		Depression		
			*χ* ^2^	*p*	Score	*χ* ^2^	*p*
Age	60~69	1309 (44.1)	17.255	<0.001	10.0 (5.0, 15.0)	4.471	0.346
	>69~79	1285 (43.3)			11.0 (6.0, 15.0)		
	>79	376 (12.7)			9.0 (5.0, 14.0)		
Gender	Male	1471 (49.5)	0.272	0.873	9.0 (5.0, 14.0)	59.772	<0.001
	Female	1499 (50.5)			11.0 (6.0, 16.0)		
Education Level	Primary school and below	2521 (84.9)	35.051	<0.001	10.0 (5.0, 15.0)	8.228	0.222
	Middle school	311 (10.5)			11.0 (6.0, 15.0)		
	High school	115 (3.9)			11.0 (8.0, 14.0)		
	University and above	23 (0.8)			10.0 (6.0., 14.0)		
Marital Status	Married	1785 (60.1)	6.068	0.048	10.0 (5.0, 14.0)	27.625	<0.001
	Unmarried	1185 (39.9)			11.0 (6.0, 16.0)		
Residence	Town	1059 (35.7)	69.413	<0.001	9.0 (5.0, 14.0)	36.905	<0.001
	Rural	1911 (64.3)			11.0 (6.0, 16.0)		
Self-rated Health	Very bad	247 (8.3)	31.575	<0.001	15.0 (10.0, 21.0)	285.138	<0.001
	Bad	822 (27.7)			12.0 (8.0, 17.0)		
	General	1419 (47.8)			9.0 (5.0, 14.0)		
	Good	289 (9.7)			6.0 (3.0, 12.0)		
	Very good	193 (6.5)			6.0 (3.0, 11.5)		
Medical Insurance	No	155 (5.2)	2.303	0.316	10.0 (6.0, 15.0)	0.486	0.784
	Yes	2815 (94.8)			10.0 (6.0, 15.0)		
Pension Insurance	No	369 (12.4)	4.155	0.125	10.0 (5.0, 16.0)	2.988	0.224
	Yes	2601 (87.6)			10.0 (6.0, 15.0)		

**Table 4 healthcare-13-01235-t004:** Correlation analysis of intergenerational care, children’s emotional support, depression and life satisfaction.

Variables	$\mathrm{Score}\;(\overset{-}{x}\;\pm \;s)$	Intergenerational Care	Children’s Emotional Support	Depression	Life Satisfaction
Intergenerational Care	0.31 ± 0.46	1	—	—	—
Children’s Emotional Support	4.43 ± 2.93	0.123 *	1	—	—
Depression	10.64 ± 6.67	−0.096 *	−0.146 *	1	—
Life Satisfaction	3.15 ± 0.68	0.141 *	0.177 *	−0.335 *	1

**Note:** * *p* < 0.001; —: omitted data.

**Table 5 healthcare-13-01235-t005:** Regression analysis of the mediating model between intergenerational care and life satisfaction through children’s emotional support and depression.

Regression Equation		Overall Fit Index			Regression Coefficient		
Outcome Variable	Predictor variable	*R*	*R* ^2^	*F*	*β* (95% *CI*)	*t*	*p*
Children’s Emotional Support	Intergenerational Care	0.206	0.043	14.586	0.210 (0.153~0.267)	7.241	<0.001
Depression	Intergenerational Care	0.365	0.133	45.376	−0.096 (−0.146~−0.047)	−3.822	<0.001
	Children’s Emotional Support				−0.095 (−0.127~−0.064)	−6.019	<0.001
Life Satisfaction	Intergenerational Care	0.413	0.17	55.152	0.162 (0.111~0.213)	6.192	<0.001
	Children’s Emotional Support				0.111 (0.079~0.143)	6.727	<0.001
	Depression				−0.288 (−0.325~−0.251)	−15.149	<0.001

**Table 6 healthcare-13-01235-t006:** Path analysis of chain mediation between intergenerational care and life satisfaction through children’s emotional support and depression.

Type of Effect	Effect	Boot*SE*	95% *CI*	Effect Size (%)
Total Effect	0.219	0.027	0.165~0.272	100.00
Direct Effect	0.162	0.026	0.111~0.213	73.97
Total Indirect Effect	0.057	0.009	0.040~0.075	26.03
Ind1 Intergenerational Care → Children’s Emotional Support → Life Satisfaction	0.023	0.005	0.015~0.033	10.50
Ind2 Intergenerational Care → Depression → Life Satisfaction	0.028	0.007	0.014~0.043	12.79
Ind3 Intergenerational Care → Children’s Emotional Support → Depression → Life Satisfaction	0.006	0.001	0.004~0.008	2.74

**Note:** Total Indirect Effect = Ind1 + Ind2 + Ind3.

**Table 7 healthcare-13-01235-t007:** Results of hypothesis testing.

Hypothesis	Supported	Test Result
H1	Yes	“Intergenerational care” positively influences “life satisfaction”
H2a	Yes	“Intergenerational care” positively influences “children’s emotional support”
H2b	Yes	“Children’s emotional support” positively influences “life satisfaction”
H2c	Yes	“Children’s emotional support” mediates the relationship between ”intergenerational care” and “life satisfaction”
H3a	Yes	“Intergenerational care” negatively influences “depression”
H3b	Yes	“Depression” negatively influences “life satisfaction”
H3c	Yes	“Depression” mediates the relationship between ”intergenerational care” and “life satisfaction”
H4	Yes	“Intergenerational care” influences “life satisfaction” through a chain mediation pathway involving “children’s emotional support” and “depression”

## Data Availability

The data used in this study were obtained from the publicly available China Health and Retirement Longitudinal Study (CHARLS) database, which is hosted by the National School of Development at Peking University. The CHARLS dataset is accessible to researchers through an application process to ensure compliance with privacy and ethical considerations. Researchers can request access to the data at http://charls.pku.edu.cn/en (accessed on 21 November 2023).

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
