# Peer review of "The Influences of Intergenerational Care on Life Satisfaction in Older Adults: Chain Mediation by Children’s Emotional Support and Depression"

_healthcare, 2025, doi:10.3390/healthcare13111235_

Round 1

Reviewer 1 Report

Comments and Suggestions for Authors

Dear Authors,

Please consider the following concrete recommendations to improve your manuscript:

  • Introduction:

    • Expand the literature review to include studies from the last five years.

    • Provide additional context on the cross-cultural differences in intergenerational care to enhance the background discussion.

  • Research Design:

    • While the cross-sectional design is appropriate for this study, explicitly acknowledge the limitations related to causal inference.

    • Discuss the potential benefits of using longitudinal data in future research to confirm the causal pathways.

  • Methods:

    • Add more details on how key variables are coded, particularly explaining the binary classification of intergenerational care.

    • Clearly describe the rationale behind your choice of statistical methods, including any assumptions made.

    • Provide a more explicit explanation of how confounding factors are identified and controlled in the analysis.

  • Results:

    • Although your tables and figures communicate the findings effectively, improve the captions and table footnotes to ensure they are self-contained and clear to all readers.

    • Explain any decision rules for categorizing responses (e.g., for children’s emotional support) directly in the table or figure legends.

  • Conclusions:

    • While the conclusions are consistent with your results, expand the discussion on policy implications.

    • Suggest concrete future research directions that could further validate the mediation model and examine related factors

Implementing these recommendations will significantly enhance your manuscript's clarity, rigor, and overall impact.

Reviewer 2 Report

Comments and Suggestions for Authors

Thank you for giving me the opportunity to review the manuscript entitled “the Influences of Intergenerational Care on Life Satisfaction in Older Adults: Chain Mediation by Children's Emotional Support and Depression”. The manuscript has been well presented. The most important concern is about measuring variables.

Please

Explain theoretically “Chain mediation model” in the introduction section

Using related theories, explain how “Chain mediation model” was developed

Provide a reference which has been previously published and described CHARLS in the method section

Provide assumptions of mediation analysis and indicate whether assumptions have been met

Discuss the findings using Confucianism instructions  

Reviewer 3 Report

Comments and Suggestions for Authors

The Influences of Intergenerational Care on Life Satisfaction in Older Adults: Chain Mediation by Children's Emotional Support and Depression
Thank you for the opportunity to review this manuscript. The authors have made a compelling, evidenced argument for the health and emotional benefits of intergenerational care (i.e., taking care of one’s own grandchildren) for older adults in the context of traditional Chinese culture. The article uses a large sample of Chinese older adults to explore the relationship between intergenerational care and life satisfaction of older adults. The article is compact but logical. There are a number of points that should be addressed. I have divided my comments into Major and minor, although they are all quite minor points, and I think the authors can resolve them easily. Congratulations on a very nice study.
Major
1. Throughout your manuscript It would be better to use the term “older adult” rather than “elderly” or “elders” according to AMA guidelines. Elderly should only be used with caution as an adjective and not a noun. This resource might be helpful: https://publichealth.wustl.edu/age-inclusive-language-are-you-using-it-in-your-writing-and-everyday-speech/ 
2. Lines 251-252: The meaning of the following sentence is not clear. Please elaborate on what you mean by “protective mechanisms” and who you mean by “Various societal participants”: “Various societal participants should be mobilized to establish protective mechanisms during the process where grandparents provide nurturing for their grandchildren.”
3. Line 258: What is meant by “increase financial support across generations”? Do you mean that children should give money to their parents to take care of the grandchildren? Is it common in China that Children do not financially support their elderly parents? It might be good to give a bit more cultural context here in addition to your discussion of Confucian filial piety ethics. I mean, discuss what the situation regarding this point is really like in contemporary China. 
4. There seems to be some repetition of ideas and redundancy between sections 4.3 and 4.4. Perhaps these could be combined, or a clearer distinction made between the two by giving more thorough description of your concepts and findings. 
5. Lines 313-323: This paragraph would be better placed into the discussion section under a heading such as “Implications”. Furthermore, it should be extended by adding descriptions and references to examples where such policies have been put in place. Have any other counties, for example, implemented targeted subsidies such as tax reductions and childcare allowances to alleviate family caregiving burdens? It would make your argument much stronger if you can find successful examples from the published literature.
6. I think you should move the limitations section (Lines: 324-333) to the end of the discussion section under a separate sub-heading “Limitations”.
7. One question that occurred to me is whether intergenerational caregiving has any additional benefits over volunteering. In one recent study from Japan, the researchers found that volunteering had a positive impact on loneliness, depression and other lifestyle factors in Japanese older adults (https://doi.org/10.3390/healthcare12212187). It would be interesting to see a discussion of whether caring for one’s own grandchildren has increased or similar impact as volunteer activities. If you could discuss this in the discussion section, I think it might be an interesting aspect.
8. As far as I can tell, no ethical approval number or date is given for this study. 
9. While the English is generally good, I think it would benefit from having a native speaker and/or expert editor read through and correct it for grammar and flow. 
Minor
Lines 29-30: Please correct this sentence: “It is essential to foster a positive and inclusive social environment should be created for intergenerational care ofolder adults.” E.g., “It is essential to create a positive and inclusive social environment for the intergenerational care of older adults.”
Line 46: “deepen” is the wrong word here. How about “increase” or “grow”?
Line 47: Punctuation problem: change the comma (,) after “older” to a dash (—) to match that before the word “evidenced” on the previous line. 
Lines 49-51: please add the country to this title “The 14th Five-Year Plan for National Economic and Social Development and the Long-Range Objectives Through 2035” e.g., “of the People's Republic of China”
Line 63 and 70. For clarity and flow, I think it would be good to use “Second,” and “Third,” instead of Additionally,” and “Lastly,”, respectively.
Line 76-78: Grammatical error: Change: “Using data from the 2018 China Health and Retirement Longitudinal Study (CHARLS) to empirically validate…” to “Data from the 2018 China Health and Retirement Longitudinal Study (CHARLS) will be used to empirically validate…”
Line 89: Please use “older adults” rather than “elderly”
Method: In some places the grammatical tense of the methods section changes from present to past. Please correct it using the past tense where appropriate and the passive voice, which is standard for methods sections. 
e.g., Line 97: Grammatical tense error: Change to: “For this study, life satisfaction was selected as the dependent…”
e.g., Lines 111, 125, 158, 159, 162: Grammatical tense error: Change “is” to “was” 
e.g., Line 143, 155, 156, 157, 160: Grammatical tense error: Change “are” to “were” 
Line 137-142 and 171 -174 : You need to add aspace between the word and the parenthesis. E.g.,: “age (in years)” and “insurance (P < 0.05)” 
Lines 155-156: Please add the manufacturer’s details in parentheses for Strata and IBM. E.g., (StataCorp LLC. College Station, Texas, USA) 
Please use “Participants” rather than “Subjects”. 
Lines 166-177, I would change the gramatical tense here to past tense: “This study included a total…”, “were involved”, “were noted”, “revealed”, etc.
Line 168: Do not use contractions: “don’t” = “do not” and put this in past tense “did not” 
Line 177, 195, 208: When referring the reader to Tables or figures try to write more specifically. E.g., “Refer to Table 1 and Table 2”, “The results of the analyses of difference are shown in Tables 1 and 2.” Unless you put them in parenthesis at the end of the corresponding sentence: “Additionally, depression significantly negatively predicts life satisfaction (β = - 0.288, P < 0.001) (Table 4).”
Table 1: Typo: “Intergenerationl” to “Intergenerational”
Tables 1 and 2: I think “Differences analysis” is better expressed as “analyses of difference”
Line 183-218: Change to past tense
Line 228: Change “selects” to “involved”
Line 238: For clarity write: “Our findings indicate…”
The authors use the word “foster” three times in the manuscript. I suggest changing at least 2 of them to synonyms as this work is commonly used by ChatGPT.

Comments on the Quality of English Language

While the English is generally good, I think it would benefit from having a native speaker and/or expert editor read through and correct it for grammar and flow. 

Round 2

Reviewer 3 Report

Comments and Suggestions for Authors

The Influences of Intergenerational Care on Life Satisfaction in Older Adults: Chain Mediation by Children's Emotional Support and Depression

Thank you for the chance to read the revised version of this manuscript. I think the authors have responded well to my comments from the first round. In Comments 5 and 6 of my first review, I suggested moving the study implications and limitations to the discussion section. While the journal Healthcare accepts free format submission, their author guidelines do suggest the following about placement of implications and limitations in the discussion section rather than the conclusions section (See: https://www.mdpi.com/journal/healthcare/instructions).

"•   Discussion: Authors should discuss the results and how they can be interpreted in perspective of previous studies and of the working hypotheses. The findings and their implications should be discussed in the broadest context possible and limitations of the work highlighted. Future research directions may also be mentioned. This section may be combined with Results.
•    Conclusions: This section is not mandatory but can be added to the manuscript if the discussion is unusually long or complex."

If you prefer to keep these in the Conclusion, I will leave that to the Editor and authors to make the final decision; however, I still believe they would be better placed in the discussion, allowing you to leave a stronger succinct conclusion.
